# Using Natural Language Processing and Social Media Data to Understand the Lived Experience of People with Fibromyalgia

**DOI:** 10.3390/healthcare12242511

**Published:** 2024-12-11

**Authors:** Lucy Bell, Beth Fordham, Sehreen Mumtaz, Reena Yaman, Lisa Balistreri, Ronald R. Butendieck, Anushka Irani

**Affiliations:** 1White Swan, Blue Fin Building, Fora, 1st Floor, 110 Southwark Street, London SE1 0SU, UK; lucy.bell@whiteswan.org.uk (L.B.); beth@whiteswan.org.uk (B.F.); 2Division of Rheumatology, Mayo Clinic Florida, Jacksonville, FL 32224, USA; mumtaz.sehreen@mayo.edu (S.M.); butendieck.ronald@mayo.edu (R.R.B.J.); 3Nuffield Department of Orthopaedics, Rheumatology and Musculoskeletal Sciences, University of Oxford, Oxford OX1 2JD, UK

**Keywords:** fibromyalgia, chronic pain, social media data, natural language processing, digital health, real world data

## Abstract

Background and Objectives: Fibromyalgia has many unmet needs relating to treatment, and the delivery of effective and evidence-based healthcare is lacking. We analyzed social media conversations to understand the patients’ perspectives on the lived experience of fibromyalgia, factors reported to trigger flares of pain, and the treatments being discussed, identifying barriers and opportunities to improve healthcare delivery. Methods: A non-interventional retrospective analysis accessed detail-rich conversations about fibromyalgia patients’ experiences with 714,000 documents, including a fibromyalgia language tag, which were curated between May 2019 and April 2021. Data were analyzed via qualitative and quantitative analyses. Results: Fibromyalgia conversations were found the most on Twitter and Reddit, and conversation trends remained stable over time. There were numerous environmental and modifiable triggers, ranging from the most frequent trigger of stress and anxiety to various foods. Arthritis and irritable bowel syndrome (IBS) were the most frequently associated comorbidities. Patients with fibromyalgia reported a wide range of symptoms, with pain being a cardinal feature. The massage, meditation and acupuncture domains were the most reported treatment modalities. Opportunities to improve healthcare delivered by medical providers were identified with current frustration relating to a lack of acknowledgement of their disease, minimization of symptoms and inadequately meeting their care needs. Conclusions: We developed a comprehensive, large-scale study which emphasizes advanced natural language processing algorithm application in real-world research design. Through the extensive encapsulation of patient perspectives, we outlined the habitual symptoms, triggers and treatment modalities which provide a durable foundation for addressing gaps in healthcare provision.

## 1. Introduction

Fibromyalgia is a common condition causing chronic, widespread pain which is often associated with sleep disturbance and fatigue and affects between 1.2 and 5.4% of the population [1]. Fibromyalgia is typically thought to represent characteristic features of nociplastic pain, a term recently introduced by the International Association for the Study of Pain (IASP) [2], defined as “pain that arises from altered nociception despite no clear evidence of actual or threatened tissue damage causing the activation of peripheral nociceptors or evidence for disease or lesion of the somatosensory system causing the pain”. Patients with fibromyalgia usually experience pain that is disproportionate to any peripheral pathology which may be present, and the pain itself is often poorly localized, deep, aching and varies in location and intensity, often unpredictably [3]. Non-pain symptoms such as intrusive fatigue, unrefreshing sleep, depressed mood and difficulty with concentration and short-term memory, alongside hypersensitivity to multiple sensory stimuli such as light, sound and light touch can be useful when diagnosing fibromyalgia [3]. Furthermore, this highlights the involvement of the central nervous system, which has been confirmed by numerous brain imaging studies showing evidence of altered connectivity between brain areas involved in pain processing, deficits in descending pain modulation, as well as dysfunctional reward processing [3]. There are a number of sociodemographic, early-life and physical factors which are thought to influence the development of nociplastic pain, but psychological factors such as depression, anxiety and catastrophizing are particularly important to consider due to the bidirectional relationship between mood and pain [3]. Recognizing that poor sleep and reduced physical activity can also be a cause or consequence of chronic pain is also important when considering an effective management plan [4]. Finally, fibromyalgia is associated with a number of other nociplastic conditions, including chronic overlapping pain conditions such as irritable bowel syndrome, pelvic floor dysfunction, chronic daily headaches, interstitial cystitis and vulvodynia, further broadening its impact on patients and extending the range of health services involved in caring for this group of patients [5,6]. Together, this complex combination of symptoms and systems involved makes effective treatment challenging to deliver.

Current evidence-based guidelines focus on prompt diagnosis, education and non-pharmacological therapies such as exercise [7,8]. Supporting patients living with nociplastic pain requires a tailored approach, utilizing shared decision making and helping patients to develop the skills they need to manage their pain [3,4,9]. Acknowledging that chronic pain is a valid diagnosis in its own right is important and can help patients feel validated, allowing them to focus on identifying strategies which work for them [4,9]. The core principles of managing nociplastic pain include providing patient education, addressing lifestyle factors, utilizing psychological therapies and treating comorbidities such as anxiety, depression and post-traumatic stress disorder [3]. All of the recommended treatment strategies can be delivered in the outpatient setting and promote self-supported management of symptoms, but barriers to implementation in clinical practice are still present. These include adequate dissemination of the latest understanding and guidance to clinicians; confusion surrounding making the diagnosis of fibromyalgia; provision of the resources needed to deliver a holistic and multidisciplinary approach and the lack of tailored treatment strategies [10,11]. Unfortunately, many patients continue to receive invasive procedures and recommendations which lack strong clinical evidence [12].

The need to bridge the gap between our current treatment paradigm and delivering effective and evidence-based healthcare for patients with fibromyalgia is becoming increasingly recognised [13,14]. Utilizing patient-driven research priorities should be at the forefront of clinical investigations to more adequately address patient concerns. For example, a research priority-setting initiative involving patients, caregivers and clinicians identified a number of research questions which are particularly relevant to patients with fibromyalgia [15]. The broad themes of the therapeutic gaps identified include the value of personalized, targeted treatment and subgrouping of patients, the efficacy of various self-management strategies and educational initiatives and identification of the ideal health care setting to provide fibromyalgia care [15]. A more recent review of the evidence on care delivery models in fibromyalgia also focused on patient perspectives of care to help identify the factors which patients valued the most [13]. Despite the laudable aims of these studies, they were inherently restricted to the perspectives of patients who volunteered and consented to take part in traditional research initiatives.

Many patients use social media for healthcare advice and information, with the most common intended use being to facilitate self-care [16,17]. Evaluating patient interactions with social media may provide the opportunity for large-scale, real-world data collection, giving insights into the lived experience which cannot be achieved through traditional research methods [18,19,20]. In turn, this can help better understand the impact of a condition, improve healthcare provision, identify gaps in the evidence base for treatments which may be effective and guide future research studies. In the field of rheumatology, previous studies involving social media have explored back pain and ankylosing spondylitis [21], the use of glucocorticoids [22] and perceptions regarding DMARD therapy [23]. There are no large-scale studies on fibromyalgia using social media data.

The aim of this study is to analyze social media conversations relating to fibromyalgia, with the goal of understanding the patients’ perspectives for this condition. Specifically, we focus on exploring how the lived experience of fibromyalgia is described on social media, capturing the factors reported to trigger flares of pain and treatments under discussion as well as identifying barriers to the delivery of optimal care and recognizing potential opportunities to improve healthcare delivery.

## 2. Materials and Methods

This is a non-interventional, retrospective analysis of social media data available in the public domain between May 2019 and April 2021. The data collection and statistical and qualitative analysis were conducted by White Swan, a registered charity in England and Wales (1176486) which aims to improve health and well-being through technology and analytics.

### 2.1. Data Collection

White Swan purchased social media posts mentioning at least one term from the following list of keywords via Twitter (now known as X) and across a range of domains from the social data aggregator Socialgist: “chronic widespread pain”, “diffuse myofascial pain syndrome”, “fibro myalgia”, “fibro mialgia”, fibromialgia”, “fibro-mialgia”, “fibromyalgia”, “fibro-myalgia”, “fibromyalgie”, “fibromyositis”, “fibrositis”, “muscular rheumatism”, myofibrositis”.

The posts from Socialgist were from social platforms such as Reddit, specialized forums such as healthunlocked.com, mumsnet.com, myfitnesspal.com, askadoctor24x7.com, babycenter.com and inspire.com and review sites such as amazon.com. These posts were accessed on 6 May 2021.

### 2.2. Taxonomy Design and Keyword and Phrase Development

The natural language processing (NLP) approach used keyword and phrase matching to detect mentions of topics of interest within a social post. These keywords and phrases were chosen to ensure comprehensive tagging of patient and healthcare professionals’ experiences.

A taxonomy structure was devised based on the project scope to provide organization of the keywords for use in the tagging process, as well as topic categorization and flexible data filtering options for the data analysis stage.

#### 2.2.1. Taxonomy Structure

The taxonomy comprised three hierarchical levels, ranging from a broad category down to the most atomic entity: the sub lens. Against each sub lens, two columns were added to capture “inclusion terms” and “exclusion terms”, as illustrated below in Table 1.

Inclusion terms are the list of keywords or phrases which were tagged in the data. Exclusion terms are keywords or phrases which, if identified in a post, would exclude the social post from being included in the dataset. This exclusion term process helped remove irrelevant content from either the overall dataset or from a specific category, lens or sub lens.

#### 2.2.2. Identifying Entities of Interest and Inclusion Terms

An iterative process was used to identify and organize a set of topics covering a wide range of fibromyalgia-related topics discussed in the data along with the language used by both medical professionals and patients. This process encompassed the following.

Expert input: A list of common symptoms, treatments, triggers and other key factors in the fibromyalgia patient journey was provided by the research team.

Desk research: A review of the topics discussed and language used in the existing literature and online patient conversations was conducted.

N-gram analysis: This involved programmatically identifying the frequency of occurrence of individual words or word sequences in the dataset. For example, 3 g analysis would output the frequencies of phrases comprising three words, such as “difficulty with sleep” and “problems sleeping before”. These N-grams were manually reviewed to identify commonly used language relating to fibromyalgia experiences.

Clinical ontology review: Clinical ontologies, including the disease ontology (DO (https://disease-ontology.org/)) and human phenotype ontology (HPO (https://www.jax.org/)), were reviewed, and relevant entities and synonyms relating to fibromyalgia and its related symptoms were identified.

Iterative social data review: Preliminary social data were reviewed to identify other keywords used by patients and healthcare professionals. The taxonomy was updated with these additional terms before the final language tagging was conducted.

Each of the additional keywords or phrases and synonyms identified through this process were assigned to the relevant points in the taxonomy structure.

The final keywords and taxonomy structure were used to tag the unstructured data.

### 2.3. Data Analysis

The following methods were used to analyze the data.

#### 2.3.1. Entity Counting

Entities were counted by the number of unique posts their tags appeared in to quantify the frequency of specific categories, lenses and sub lenses. These counts were analyzed to provide insights into the prevalence and relevance of key concepts within the dataset.

#### 2.3.2. Entity Associations

Identifying the associations between entity pairs involved measuring the number of social data posts in which each entity had at least one mention, which allowed us to highlight pairs of co-occurring entities which appeared to be unusual or interesting and then filter the tagged dataset to posts where they were mentioned together for further qualitative or quantitative analysis. Instead of simply relying on the entity pair co-occurrence volumes, associations were identified using their normalized pointwise mutual information (NPMI), which effectively measured the strength of their association while accounting for statistical significance.

NPMI is a continuous value which ranges from −1 to 1 with the following meanings:

NPMI = −1: The entity pair has no co-occurring tags.

−1 < NPMI ≤ 0: The entity pair has co-occurring tags, but the relationship between them is not considered statistically significant, and the entities are statistically independent of each other.

0 < NPMI ≤ 1: The entity pair has co-occurring tags, and the relationship is considered statistically significant.

1: The tags co-occur perfectly.

NPMI is calculated using the following formula:$$NPMI\left(e_{1};e_{2}\right)=\frac{PMI\left(e_{1};e_{2}\right)}{-\log p\left(e_{1},e_{2}\right)}$$
where
$$PMI\left(e_{1};e_{2}\right)=\log\frac{p\left(e_{1},e_{2}\right)}{p\left(e_{1}\right)p\left(e_{2}\right)}$$

In the equations above, the probability of entity x, $p\left(e_{x}\right)$, is the relative frequency of unique documents mentioning entity x in the dataset, and the joint probability $p\left(e_{x},e_{y}\right)$ is the relative frequency of unique documents which mention both entities.

A final stage of analysis reduced the recorded significance the further away the two measured entities appeared in the text. For example, entities denoting fibromyalgia two paragraphs away from an entity denoting symptom pain would result in a less significant NPMI score than if the two denotations appeared in the same sentence. The volume of identified symptoms, co-morbidities and triggers and the associated NLP-adjusted NMPI score informed the analysis of fibromyalgia symptoms.

#### 2.3.3. Qualitative Analysis

The qualitative analysis involved reading through many social data posts relating to the key concepts identified in the quantitative analysis. Thematic analysis was conducted to verify the robustness of taxonomy groupings, and contextual analysis was used to understand the wider context and identify illustrative examples to support the quantitative findings.

### 2.4. Summary of Data Tools

The parameters for data collection (timeframe, data sources and keywords) were configured through a proprietary data purchase tool which connected to the data providers’ systems via their APIs, which enabled programmatic querying and ingestion of data.

Taxonomy development and iterative updates to keywords were managed using a proprietary taxonomy management system.

An in-house data tagging and statistical reporting pipeline tool running on a distributed computing cluster was used to tag the dataset with entity mentions and taxonomy levels and provide reports such as topic volumes and association metrics.

A dataset explorer tool allowing the researchers to segment and view the social posts based on keywords, domains and taxonomy level tags was used for qualitative analysis.

## 3. Results

In total, 714,000 documents which included a fibromyalgia language tag were identified between May 2019 and April 2021 (Figure 1).

### 3.1. Benchmarking Conversations over Time

Data were drawn from a combination of long- and short-form posts from a variety of sources, but the majority were from Twitter (82%) and Reddit (14%). This provided a useful mix of high-volume trends from Twitter and rich, long-form discussion from Reddit which provided context for the trends. Similar distributions were seen in other comparator conditions associated with pain, including irritable bowel syndrome and arthritis (Figure 2).

Fibromyalgia conversations were relatively constant during the study period. There was an increase in the number of conversations occuring in March 2020, which was atrributed to COVID-related conversations. This is supported by a similar increase in frequency seen in conversations linked to all medical conditions (Figure 3).

### 3.2. Patient Experience of Symptoms Associated with Fibromyalgia

Patients with fibromyalgia reported a wide range of symptoms, which could be grouped into six key areas: pain, fatigue and sleep disturbance, mental health, unexpected sensations, cognition and sense sensitivities (Figure 4). Pain, also considered to be the cardinal feature of fibromyalgia by healthcare providers, was the most prominent and frequently discussed symptom. Unexpected sensations comprised uncomfortable and confusing sensations in the muscles and skin, which could include inflammation, swelling, stiffness, numbness and cramping. In addition, patients reported experiencing muscle spasms and restless legs. Sense sensitivities described heightened sensitivity to environmental factors, most frequently touch, light, noise and weather, which caused immediate pain and symptom flares.

### 3.3. Factors Triggering Symptom Flares

Fibromyalgia symptom flares were reportedly triggered by various factors, with stress and anxiety being the most common symptoms (Figure 5). Patients frequently reported that emotional stress or anxious situations could lead to intensified pain and fatigue. Weather, temperature and air quality were also significant triggers. Many individuals experienced flares in response to changes in climate, such as rainy or chilly weather, extreme heat, high humidity and increased air pollution. These environmental factors could exacerbate symptoms, making day-to-day activities more challenging.

Other triggers included certain foods and vaccines, though these were discussed less frequently. Patients reported still exploring which specific foods might provoke flares, and while vaccines were reported to trigger symptoms, there was no evidence of strong anti-vaccine sentiment within the fibromyalgia community. COVID-19 infection was also linked to flare-ups, although patients often struggled to distinguish whether their symptoms stemmed from fibromyalgia, vaccine side effects or the virus itself. Physical activity was a known trigger, but it garnered less discussion, as many patients have learned to manage their activity levels to avoid flares. Additionally, hormonal changes, alcohol and even sensitivity to bright sunlight have been cited as possible contributors to symptom exacerbation.

Food and drink sensitivities played a significant role in triggering fibromyalgia symptom flares, with gluten and wheat being among the most reported culprits (Figure 6). Many fibromyalgia patients also found that sugar, dairy and lactose could worsen their symptoms, particularly increasing pain and fatigue. Carbohydrates and keto diets could be problematic, as some patients struggled to balance their intake to avoid flares. Alcohol was discussed as another common trigger, while caffeine posed a unique challenge. Although patients often relied on caffeinated beverages to manage fatigue, caffeine itself could trigger symptom exacerbation, creating a difficult cycle for those seeking relief.

Lesser-known triggers, such as histamines and nightshade vegetables, received less attention, though they were also listed as potential triggers for fibromyalgia symptoms. Patients were generally more aware of the impacts of gluten, sugar and dairy, as alternatives for these food groups are widely available in stores. However, options for those sensitive to histamines and nightshades were limited, making dietary management more challenging. Additionally, artificial sweeteners, soy and processed foods have been cited as potential triggers, further complicating food choices for fibromyalgia sufferers looking to avoid symptom flares.

### 3.4. Beneficial Nutrients

Fibromyalgia patients often discuss the benefits of incorporating specific nutrients into their diets to help manage symptoms and reduce the severity of flares. Many reported that maintaining adequate vitamin D levels, often through supplements, plays a crucial role in managing their condition. Patients frequently rely on medical tests to monitor their vitamin D levels and adjust supplementation as needed. In addition to vitamin D, magnesium is commonly used to alleviate symptoms, with patients noting that magnesium supplements and magnesium baths can help ease muscle pain and reduce flare intensity. By addressing nutrient deficiencies, patients feel better equipped to manage the unpredictability of fibromyalgia flares.

### 3.5. Common Comorbidities

Arthritis and irritable bowel syndrome (IBS) were the most common associated comorbidities which were identified, with 40,916 and 16,663 mentions, respectively. Although fibromyalgia and arthritis are thought of as distinct conditions, both can cause significant debilitating pain and discomfort. This can adversely affect the chances of success in pain and discomfort reduction by inhibiting the ability to exercise and lose weight. Both fibromyalgia and IBS can be triggered by dietary choices, and flares of fibromyalgia and IBS can occur simultaneously. Despite this finding, there were few conversations about dietary triggers associated with both fibromyalgia and IBS flares.

### 3.6. Treatment and Support Available

Massage therapy wass identified as the most frequently discussed treatment with 7009 mentions. Social media statements described regular massages as a beneficial treatment, with alleviation of pain and “loosening” of muscles. The addition of heat to massages, such as in the form of hot stones, was also discussed as being helpful. The limitations described included massage therapist skill level, access, especially during the COVID-19 pandemic, leading to physical distancing recommendations, and the duration of benefit, with discussions describing this as a short-term solution without perceived benefits after 6 months.

Other discussed therapies in decreasing order of online mentions included meditation, acupuncture, and mindfulness, with 5090, 3778, and 2936 mentions, respectively. Mentioned medications included non-steroidal anti-inflammatory drugs, amitriptyline, anti-epileptic drugs, cyclobenzaprine and selective serotonin reuptake inhibitors, with mention counts ranging from 1154 to 2301. Cognitive behavioral therapy received a total of 1051 mentions. The identified treatments, listed by decreasing number of mentions, are summarized in Table 2.

The term “support” was mentioned a total of 117,533 times. Discussions regarding medical visits and appointments described fibromyalgia patients often meeting with more than one and up to five medical professionals regarding their condition. The most frequently discussed medical professionals were psychotherapists and therapists, with 7713 and 7081 mentions, respectively. These visits were described as targeting the effects of mental health factors, such as stress, on fibromyalgia flares, as well as the mental health effects of pain. Similarly, pain psychologists were discussed in relation to treatment for chronic pain. Dietary changes were noted as a natural treatment for fibromyalgia, with “dietician” receiving 234 mentions. Chiropractors were mentioned 1967 times with themes related to consultation for the symptom of severe back pain.

Themes of frustration with medical providers, including physicians, were related to lack of acknowledgement of their disease, minimization of symptoms and inadequately meeting their care needs. Mental health and medical professionals were also described as the professionals sought prior to establishing the correct diagnosis for their symptoms. The identified support professionals, listed in order of decreasing number of mentions, are summarized in Table 2.

Peer support groups were viewed positively and perceived as a nonjudgemental forum to discuss pain and emotions. Individuals with fibromyalgia described membership in multiple support groups, including those for other comorbidities such as insomnia, and making friends with others in support groups. The feeling of support groups being draining was described as a downside. These were also considered to be more understanding audiences compared with those perceived as not considering fibromyalgia to be a serious condition, such as family members and medical providers. Family was mentioned 91,369 times, with varying descriptions including gratitude for family support, guilt related to inability to spend time with family and frustration regarding a lack of understanding of patients’ symptoms and the impact of their disease. Similarly, varying sentiments regarding spirituality were identified. This included anger, considering their disease to be punishment, viewing their disease as a challenge of their faith and finding faith and spirituality to be an avenue for acceptance of their diagnosis. “Spiritual” was mentioned 1706 times. “Emotional” was mentioned 8990 times, with descriptions of emotional responses related to diagnosis, physical or emotional trauma causing their diseases and anger at themselves or their bodies.

## 4. Discussion

This is the first large-scale study using social media data to utilize natural language processing to investigate the patient perspective on living with fibromyalgia. While the majority of the symptoms identified by patients associated with fibromyalgia were reassuringly consistent with those recognized to be part of the condition and included in the latest diagnostic criteria [24], some additional symptoms were highlighted by the patient voice. These include sensations in the skin described as inflammation, swelling, stiffness, numbness and cramping as well as muscle-related features, including spasms and restless legs. Sensitivity to non-nociceptive stimuli such as light, noise and weather was identified by patients, reminding clinicians to elicit these positive findings—which are typical for nociplastic pain—when evaluating patients [3]. The most frequently discussed triggers for fibromyalgia flares were stress and anxiety, weather-related factors and food. This information is useful to share with patients to help them identify and potentially mitigate triggers they may not have recognized yet. Furthermore, this information provides guidance to researchers regarding priority areas for investigation. Stress and anxiety have been extensively studied in relation to fibromyalgia and chronic pain more widely, but the study of weather-related factors is an emerging area [25]. While there has been great interest in the effect of diet on fibromyalgia, the current evidence base remains inconclusive. Food-related allergies may be more frequent than in the general population [26], but this may represent hypersensitivity rather than a traditional form of allergy, being similar to drug hypersensitivity in this patient group [9]. Further work in this area is clearly needed.

Fibromyalgia treatment options discussed by patients via social media overall were remarkably aligned with the current guidance and evidence base [3,9,27,28]. However, an apparent discrepancy was identified in relation to the role of massage therapy. The current study identified that massage therapy wass the most discussed treatment modality among patients, but this is not endorsed by any current therapeutic guidelines. Several meta-analyses on limited original studies have shown the benefits of massage therapy, although these differ by massage style and the symptom of interest [29,30]. Together, this provides justification for the pursuit of high-quality studies investigating the effectiveness of massages in this condition. In the interim, these data, providers’ familiarization with complementary or alternative medicine interventions as well as available information about their potential benefits may aid in having meaningful conversations regarding options which patients appear to frequently consider or pursue.

These data also provide invaluable insights into the patient perspective on navigating the healthcare system and sources of support available to them. The involvement of multiple different medical professionals was identified alongside frustration with medical providers due to lack of acknowledgement of their disease and its impact, in addition to not being able to adequately meet their care needs. Effective communication is essential to ensuring that people living with fibromyalgia feel listened to, validated, supported and empowered as they develop strategies to help improve their quality of life [4,9]. There is a move toward increasing awareness amongst healthcare providers through broad messaging [4,9,31] as well as by addressing specialty specific audiences [3], but this education needs to continue to grow and develop as broad a reach as possible. However, peer support groups were considered to be helpful compared with family members and medical providers who do not necessarily share the lived experience of chronic pain, but they can also be draining, which is an important potential limitation. A smaller-scale study examining social media engagement found that while many patients with fibromyalgia use social media platforms as a source of community, emotional support and information about their conditions, the frequency and type of social media use can have varying effects on their well-being [32]. Increased social media use, especially for information-seeking or engaging in discussions about symptoms, can lead to heightened symptom awareness, which may in turn increase self-reporting of symptom severity and exacerbate symptoms like pain, fatigue and cognitive issues [32]. Based on these data, healthcare providers should consider how they may be able to help patients find suitable peer support opportunities. Different sentiments in relation to spirituality were identified, which serves as a reminder to clinicians to be open to supporting patients as they navigate these emotions in a patient-centric manner.

A recent study examining tweets in English and Spanish on X regarding chronic pain conditions was published [33]. Following paraplegia, headache and fibromyalgia ranked second and third in terms of number of tweets. Topics such as symptoms, medical treatment and association with negative emotions were catalogued. Interestingly, their group identified a sharp increase in fibromyalgia-related tweets in the second quarter of 2019, with a subsequent decrease in the second quarter of 2020. This contrasts with our study, which identified the highest number of mentions in March and April of 2020. These findings have been attributed to the effects of the COVID-19 pandemic during this period, but the inverse trend may be attributable to the different datasets, inclusion criteria or geographic distribution. This study further demonstrates the feasibility of social media-based studies in providing a cross-sectional evaluation of patient perspectives.

The main strength of this study is the ability to analyze large volumes of data drawn from a variety of social media sources over a 2 year period. The findings were limited to online discourse taking place in English during the specified timeframe. While the current study design aids in eliminating nonresponse bias attributable to an inability or unwillingness to participate in traditional research studies, it remains limited to the view of those able and willing to discuss these topics online without the ability to link their medical diagnoses to formal medical records. This therefore limits the generalizability of the current findings to the population of patients with a verified clinical diagnosis of fibromyalgia. Furthermore, there may be significant bias in terms of the age and sex of people who are willing to discuss and share their health-related experiences online. Combining this information with data obtained using more traditional research methods may provide the most comprehensive, inclusive and generalizable approach to capturing the perspectives of this patient population.

From a practical perspective, there are three main ways that this study can assist with the delivery of high-quality care required for this complex, multifactorial condition. First, sharing the results with patients and healthcare providers serves an important role in increasing awareness of common triggers experienced by people living with fibromyalgia. This can help make patients feel validated and reassured that their experience is not unique or occurring in isolation. Second, the commonly reported modifiable symptom triggers, including stress, anxiety and food, can be highlighted during consultations as areas which may provide additional targets for treatment. This can be coupled with the most discussed treatment strategies—massage, meditation, acupuncture and mindfulness—as guidance for ways to address the mental health aspect of the condition. Acknowledging the likely role of dietary factors can also lead to further engagement with integrative medicine or allergy approaches to explore the role of nutrition. Third, the areas of discrepancy between topics being frequently discussed and the current evidence base should be addressed in future research focusing on the role of, for example, diet and massage therapy.

In summary, the current study demonstrates the utility of applying natural language processing algorithms to large-scale, real-world data to capture the broader patient perspective of living with fibromyalgia. As evidenced by these data, this approach enabled us to improve our understanding of the collection of symptoms attributable to fibromyalgia, identified areas of priority for future research into therapeutic options and drew attention to gaps in healthcare provision through the lens of those with experience living with fibromyalgia.

## Figures and Tables

**Figure 1 healthcare-12-02511-f001:**
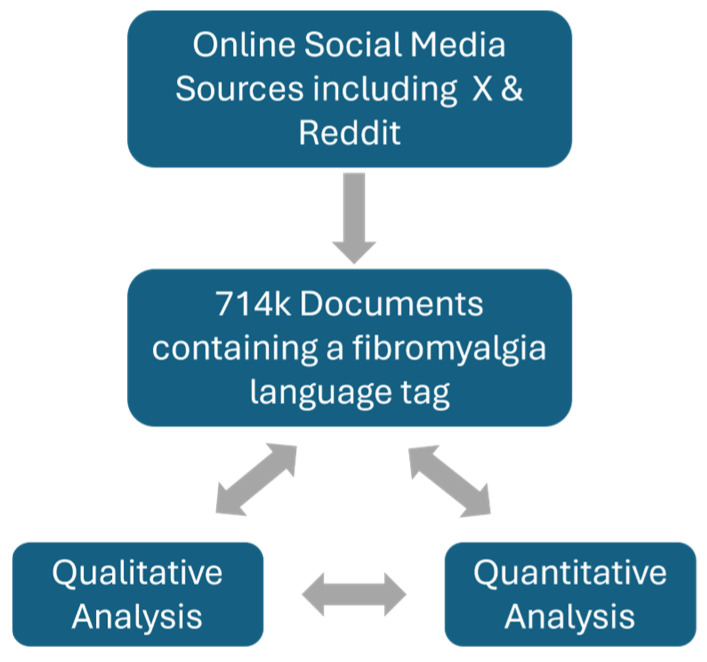
Study flowchart. In total, 714,000 relevant documents including a fibromyalgia language tag were identified and used for subsequent qualitative and quantitative analyses.

**Figure 2 healthcare-12-02511-f002:**
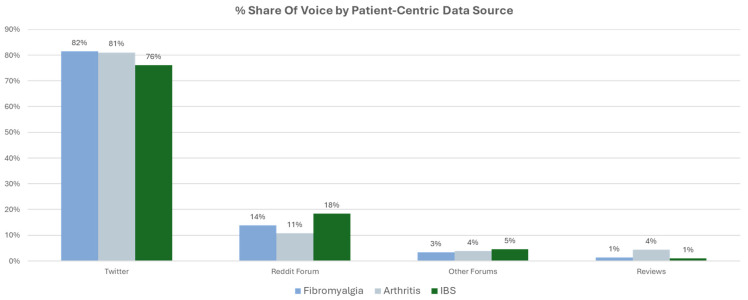
Data source distribution across platforms for fibromyalgia, arthritis and irritable bowel syndrome (IBS).

**Figure 3 healthcare-12-02511-f003:**
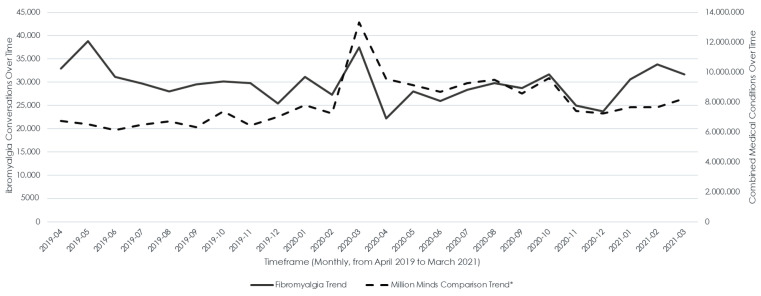
Trend of fibromyalgia conversations over time compared with all medical conversations. * Data from Million Minds database, comprising 600 medical conditions.

**Figure 4 healthcare-12-02511-f004:**
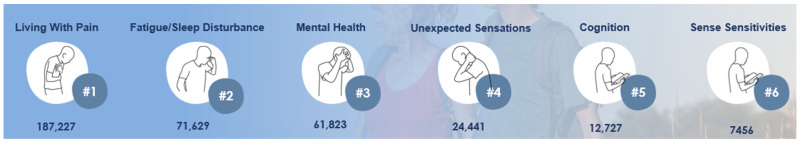
Symptoms of fibromyalgia in order of frequency of appearance in conversations out of 263,840 total global conversations.

**Figure 5 healthcare-12-02511-f005:**
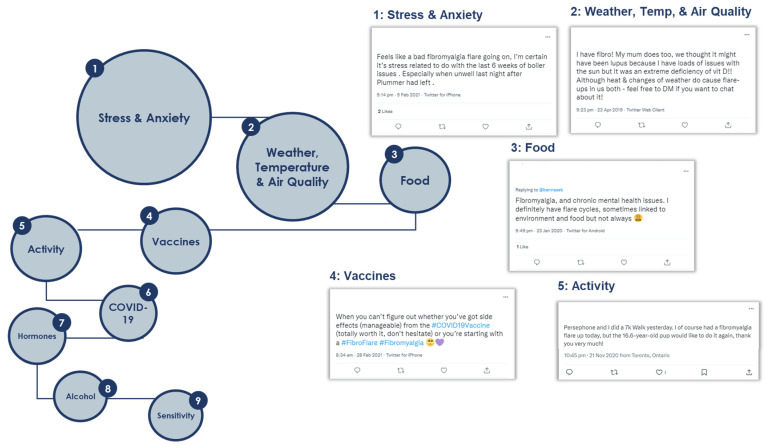
Commonly discussed triggers for flares. (The size of the circle is representative of the frequency of discussions around the topic.)

**Figure 6 healthcare-12-02511-f006:**
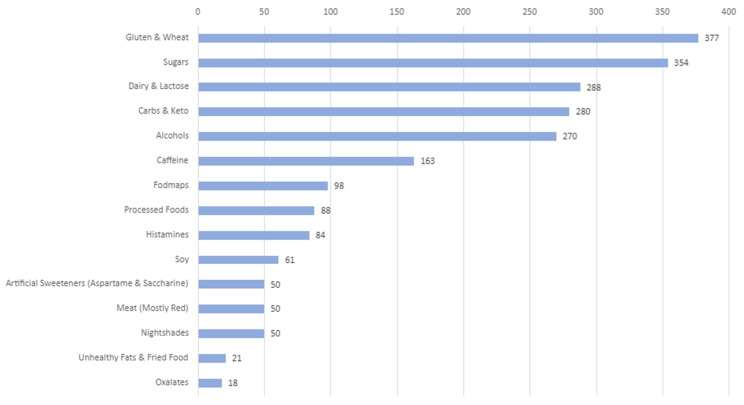
Most conversed about trigger foods and drinks.

**Table 1 healthcare-12-02511-t001:** Example of taxonomy structure applied.

1. Category	2. Lens	3. Sub Lens	4. Inclusion Terms	5. Exclusion Terms
Symptom Triggers	Ingredient Triggers	Histamines	Histamine/histamineintolerance/histamines	Gardening/horticulture/my garden

**Table 2 healthcare-12-02511-t002:** The range and number of mentions of treatment options, forms of provider support and other domains being discussed.

	Treatment	Number of Mentions
Therapies	Massage	7009
	Meditation	5090
	Acupuncture	3778
	Mindfulness	2936
	Nonsteroidal anti-inflammatory drug	2301
	Amitriptyline	1948
	Anticonvulsant	1282
	Cyclobenzaprine	1178
	Selective serotonin reuptake inhibitor	1154
	Cognitive behavioral therapy	1051
Providers	Psychotherapist	7713
	Therapist	7081
	Neurologist	4084
	Health professional	3605
	Psychiatrist	3474
	Support group	3077
	Psychologist	2604
	Chiropractor	1967
	Dietician	234
	Radiologist	110
Other Domains or People	Support	117,533
	Family	91,369
	Doctor	72,740
	Medical	40,915
	Emotional	8990
	Other sufferers	4428
	Financial	3909
	Practical	1900
	Spiritual	1706

## Data Availability

Raw transaction data were not made available, as their ownership is retained by White Swan (www.whiteswan.org.uk, accessed on 6 May 2021).

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
