# Peer review of "Using Natural Language Processing and Social Media Data to Understand the Lived Experience of People with Fibromyalgia"

_healthcare, 2024, doi:10.3390/healthcare12242511_

Round 1
Reviewer 1 Report
Comments and Suggestions for Authors
1)The manuscript could benefit from more explicit comparisons with existing fibromyalgia literature to contextualize these social media-derived insights. Also, expanding the introduction to include a broader background on fibromyalgia management and patient challenges would strengthen the context for readers unfamiliar with the condition.
2)Consider including evidence on the psychological impact of fibromyalgia (see 10.1093/pm/pnad139 ; https://doi.org/10.2147/PRBM.S178240) , as well as the contributing factors to its development, particularly central sensitization (see https://doi.org/10.1016/j.semarthrit.2014.01.001) and psychological factors. This addition would provide a more comprehensive view of the condition's multifaceted nature, emphasizing the role of both neurological and psychological elements in shaping the patient experience.
3)I would suggest adding psychological treatments in addition to exercise as a non pharmacological treatment. Incorporating therapies like cognitive-behavioral therapy (CBT) (see 10.1016/j.ijchp.2020.04.002), ACT or mindfulness-based interventions (see 10.1007/s11926-017-0686-0) could address both the mental health challenges and pain management needs in fibromyalgia, providing a more holistic approach to patient care.
4)While the methods are robust, the paper could benefit from additional clarification on potential limitations in social media data reliability (e.g., demographic biases, non-verified self-reports). Additionally, the NLP and taxonomy-based segmentation processes could be further detailed to enhance reproducibility.
5)Consider discussing more explicitly how these insights might translate into practical clinical interventions, especially concerning mental health and lifestyle support. Additionally, limitations of the study, such as the absence of non-English data and inability to verify medical diagnoses, should be emphasized to give a balanced view.
Author Response
1)The manuscript could benefit from more explicit comparisons with existing fibromyalgia literature to contextualize these social media-derived insights. Also, expanding the introduction to include a broader background on fibromyalgia management and patient challenges would strengthen the context for readers unfamiliar with the condition.
Thank you for this comment, we agree and have expanded the introduction to provide a more comprehensive overview of the condition and current challenges for the reader (first 2 paragraphs of the introduction).
2)Consider including evidence on the psychological impact of fibromyalgia (see 10.1093/pm/pnad139 ; https://doi.org/10.2147/PRBM.S178240) , as well as the contributing factors to its development, particularly central sensitization (see https://doi.org/10.1016/j.semarthrit.2014.01.001) and psychological factors (see 10.2147/JPR.S370718 ; 10.3390/jcm10040803). This addition would provide a more comprehensive view of the condition's multifaceted nature, emphasizing the role of both neurological and psychological elements in shaping the patient experience.
Thank you for highlighting this important aspect of fibromyalgia. We have included further reference to the role of psychological and other factors in the development of nociplastic pain (see lines 38-58).
3)I would suggest adding psychological treatments in addition to exercise as a non pharmacological treatment. Incorporating therapies like cognitive-behavioral therapy (CBT) (see 10.1016/j.ijchp.2020.04.002), ACT (see 10.55563/clinexprheumatol/7hvaya) or mindfulness-based interventions (see 10.1007/s11926-017-0686-0) could address both the mental health challenges and pain management needs in fibromyalgia, providing a more holistic approach to patient care.
We agree with this point and have included a more comprehensive discussion of the current evidence-based approach to supporting patients with fibromyalgia (see lines 66-73).
4)While the methods are robust, the paper could benefit from additional clarification on potential limitations in social media data reliability (e.g., demographic biases, non-verified self-reports). Additionally, the NLP and taxonomy-based segmentation processes could be further detailed to enhance reproducibility.
Thank you for this point, which was also raised by reviewers 2 and 3. We have provided further detail regarding the methods used for this analysis (see lines 163 -272). The limitations have been expanded too (see lines 502-508).
5)Consider discussing more explicitly how these insights might translate into practical clinical interventions, especially concerning mental health and lifestyle support. Additionally, limitations of the study, such as the absence of non-English data and inability to verify medical diagnoses, should be emphasized to give a balanced view.
Thank you for highlighting this. We have expanded the limitations section (see lines 502 – 508) and also given more clarity regarding the translation to practical changes in treatment approaches (see lines 510-523).
Reviewer 2 Report
Comments and Suggestions for Authors
This study provides insights to understand the experiences of patients with fibromyalgia using social media data and AI. However, there are some areas that need clarification.
Major comments:
1. Some critical details are missing in the data source and analysis section. The authors should provide a detailed description of the data collection process for each social media platform (Twitter, Reddit, etc.), including information on any APIs or tools used for data extraction. Also, please explain any search terms or filters used to identify relevant posts.
2. The study acknowledges limitations related to the online nature of the data. However, it is also important to address the potential generalizability of the findings to the fibromyalgia population. How might the demographics of people who discuss fibromyalgia online differ from those who do not participate in such online discussions?
3. The description of the ethical review is incomplete. It is important to provide a more detailed justification for these exemptions.
Author Response
- Some critical details are missing in the data source and analysis section. The authors should provide a detailed description of the data collection process for each social media platform (Twitter, Reddit, etc.), including information on any APIs or tools used for data extraction. Also, please explain any search terms or filters used to identify relevant posts.
Thank you for highlighting this point, which was also raised by the other reviewers. We have provided further detail regarding the methods used for this analysis (see lines 163 -272).
- The study acknowledges limitations related to the online nature of the data. However, it is also important to address the potential generalizability of the findings to the fibromyalgia population. How might the demographics of people who discuss fibromyalgia online differ from those who do not participate in such online discussions?
Thank you for this point, we have expanded the limitations section to address this (see lines 502-508).
- The description of the ethical review is incomplete. It is important to provide a more detailed justification for these exemptions.
Thank you for identifying this omission. This has been addressed in lines 547-8.
Reviewer 3 Report
Comments and Suggestions for Authors
1. First of all, the abstract and introduction sections of the study should definitely be rewritten. These sections are far from preparing a basis for the study and providing information about the study.
2. There is almost no information in the text about how the study was conducted and unfortunately there is very poor technical information.
3. What is White Swan?
4. Who are White Swan volunteers?
5. How were the statistics for this data obtained? Which programs were used?
6. It is stated that artificial intelligence was used in the study. But what is written is just an interpretation of statistics. How was artificial intelligence used?
Comments on the Quality of English Language
The English could be improved to more clearly express the research.
Author Response
- First of all, the abstract and introduction sections of the study should definitely be rewritten. These sections are far from preparing a basis for the study and providing information about the study.
Thank you for this feedback. We have updated these sections.
- There is almost no information in the text about how the study was conducted and unfortunately there is very poor technical information.
Thank you for this point, which was also raised by reviewers 1 and 2. We have provided further detail regarding the methods used for this analysis (see lines 163 -272)
- What is White Swan?
Thank you for this query. We have provided further information in lines 165-7.
- Who are White Swan volunteers?
Thank you for this query. We have provided further clarification regarding the input from White Swan volunteers in the acknowledgement section (lines 556-560).
- How were the statistics for this data obtained? Which programs were used?
Thank you for this comment, we have included this in the methods section (lines 262-272).
- It is stated that artificial intelligence was used in the study. But what is written is just an interpretation of statistics. How was artificial intelligence used?
Thank you for this point. We have updated the title, keywords and abstract to reflect the use of Natural Language processing which was used in the study.
Round 2
Reviewer 2 Report
Comments and Suggestions for Authors
I would like to express my appreciation to the authors for their response to the previous review comments. I confirm that the manuscript has been substantially improved and have no further comments at this stage.